# The Effect of Grandparenting on the Depression and Life Satisfaction among Middle-Aged and Older Chinese Adults

**DOI:** 10.3390/ijerph191710790

**Published:** 2022-08-30

**Authors:** Lijuan Chen, Yiang Li, Qiuyue Yang

**Affiliations:** 1The High-Quality Development Evaluation Institute, Nanjing University of Posts and Telecommunications, Nanjing 210003, China; 2Division of the Social Sciences, University of Chicago, Chicago, IL 60637, USA; 3School of Social and Behavioral Sciences, Nanjing University, Nanjing 210023, China

**Keywords:** grandparenting, well-being, life satisfaction, older adults, China

## Abstract

Given the prevalence of depressive mental health symptoms among Chinese adults of grandparenting age in recent decades, a better understanding of how depression and life satisfaction among middle-aged and older adults in China are affected by their role as grandparents is called for. This study examines the relationship between grandparenting and depression and life satisfaction among Chinese adults using multilevel regression models based on a multilevel matching dataset formulated from the 2018 *China Health and Retirement Longitudinal Study* (CHARLS) and the 2018 *China City Statistical Yearbook*. The results show that for adults who take care of their grandchildren, living with their children can significantly reduce depression. Meanwhile, whereas spending more time taking care of grandchildren can lower life satisfaction, taking care of more grandchildren is related to higher life satisfaction. The findings of this study should help policymakers improve the quality of life of Chinese adults through better-targeted approaches.

## 1. Introduction

Following the trend toward rapidly increasing life expectancies [1], China’s old-age dependency ratio is expected to rise from 17.7 in 2019 to 27.4 in 2030 [2]. In response to the rapid aging of the population, China abolished the one-child policy, which had restrained couples from having more than one child, in 2016 [3] and allowed couples to have up to three children in 2021 [4]. Early estimates suggest that this long-term policy will not lead to evident improvements in the population structure for at least two decades [5]. Meanwhile, a significant lacuna in the understanding of the subjective well-being of older Chinese adults remains. Specifically, as most Chinese people provide grandparental care, including co-residential grandparenting, in the latter period of their lives [6], the prevalence of depressive mental health symptoms and low life satisfaction during their prolonged grandparenthood has become a compelling issue in the domain of public health and social policy, necessitating a better understanding of how the grandparental role of adults in China affects their subjective well-being [7].

In view of the cultural differences in the intergenerational relationship between East and West, grandparenthood for Chinese adults may affect their well-being differently than is the case for adults from other cultural contexts. On the one hand, grandparenting is often seen as not only a blessing but also a challenge in the West. Among American adults, grandparenting was found to elevate the level of psychological stress [8], and regular provision of grandparental care was found to be negatively associated with the number of social activities in which the older adults participated [9]. On the other hand, grandparents in Asia seem to perceive and experience grandparenthood differently. Grandparents in China, on average, spend 23.09 h per week caring for their grandchildren, a level roughly equivalent to the time devoted to caring for them by their mothers [10]. The reasons it is so prevalent for grandparents to play a major role in caring for their grandchildren are, however, down to China’s unique historical norms, socioeconomic development, and demographic changes. The Chinese Confucius doctrine has led to intergenerational relationships that are primarily in solidarity and within extended families [11], and grandparents ultimately become an intrinsic aspect of the Chinese family structure [12]. In ancient Chinese history, grandparenting was considered a right and expectation by adults [13]. The patriarchal extended family was historically regarded as the ideal family type in Chinese culture [14], with sizes and compositions that varied by region. In contemporary China, the three-generation family structure remains prominent among the general population [15], and the traditional co-residence pattern persists despite the trend of modernization [16,17]. Grandparental care in China also takes place while not co-residing. Over 60% of older aged over 35 live in the same city/county but separate households with their adult children’s families, most of whom live in the immediate neighborhood [18,19]. Unlike their Western counterparts, Chinese grandparents were found to suffer no health deficits (Chen & Liu, 2012), and grandmothers who provided long-term care in a similar Asian culture, South Korea, were found to have better life satisfaction and self-rated health than those not engaged in grandparenting [20].

### 1.1. Theoretical Framework

Social exchange theory is often utilized in social gerontology to explain rational decision-making behind intergenerational support, such as grandparental care, in which people have higher life satisfaction when receiving more support than support provided [21,22]. The key underlying assumptions are that the various actors each bring material and non-material resources; the exchange is reciprocal, and it continues so long as the benefits of receiving support outweigh the costs [23]. In contemporary China, the high costs of manual labor and inadequacy of social childrearing make grandparents the cheapest as well as the optimal option for working parents [24]. Meanwhile, the Chinese seniors find the pension system provided by their employers or the welfare state inadequate, and themselves still dependent on support from their adult children [15]. Thus, the social exchanges through grandparental care within extended families become the “life-boat” for older people under economic uncertainty and instability [25]. Studies have revealed the positive effects of co-residential care among Chinese older people on improving life satisfaction and depressive symptoms [26].

Although the social exchange theory renders insights into the overarching context of intergenerational support by leading the discussion on grandparental care, it tends to overlook the nuances in grandparenthood by the care intensities that cannot be compared against other material gains. From the perspective of role theory, grandparental care may have a positive impact on subjective well-being. As outlined in the “role enhancement” theory, having multiple roles could lead to better well-being as individuals perceive a sense of competence and control in their later life [27]. Through the provision of grandparental care, the emotional rewards and intergenerational gratification may positively affect their subjective well-being [28].

Conversely, the “role strain” postulates that one’s mental health status and life satisfaction will be negatively affected when the caring obligations exceed their coping capacity [29]. Grandparenthood could be a counter-transition because it is a life change brought about by a child’s transition to parenthood [30]. Older adults tend to have little agency in deciding either the marital status of their adult children or the timing of when they will become grandparents and are sometimes required to entirely restructure their intergenerational relationships [31], which will affect their well-being on three core psychological, social, and physical fronts. Hence, the older people may have worse health when the economic pressure brought by the internal migration in contemporary Chinese society alongside its Confucius historical heritage has compelled grandparents to become “family maximizers” [32], sacrificing their own time and resources for the intensive care of their adult children and the entire family [33,34]. Apart from the provision of supplementary care, the grandparents of the left-behind children are sometimes required to take on a primary or even custodial care role that may have even worse health status [35].

### 1.2. The Effect of Grandparenting on Depression

Grandparenting is associated with both financial and opportunity costs. When the level of financial cost is moderate, the well-being of grandparents may not suffer. Among Mexican Americans, grandparents were found to have fewer depressive symptoms if they provided only a moderate level of financial help [36]. In Chile, a country of similar cultural background to Mexico, a gendered outcome regarding the costs of grandparenting was reported: the mental health status of Chilean grandfathers improved when providing financial assistance to grandchildren, but the opposite was the case for Chilean grandmothers, whose satisfaction was negatively associated with the provision of time and monetary assistance [37].

However, for both genders, in most cultural contexts, when the support required keeps increasing, their depressive symptoms will begin to deteriorate, as some understand that the excessive demands placed on the grandparents are felt to violate the inter-generational distribution of responsibilities [38,39]. Such is especially the case with regard to custodial grandparenting, where the costs are particularly high, and grandparents are often exclusively responsible for the provision of financial support to the grandchildren [40]. In mental health assessments, they have also been found to have a higher probability of depressive symptoms [41]. In an aging society where everyone works for a longer period of their life, this may only occur more frequently in the future. Sometimes, grandparents may also become the recipients of financial support for childcare, which has been found to have no association with their well-being [36]. The reason may be that the compensation provided by their adult children is not sufficient to offset much of their grandparenting expenses. Regular grandparenting also takes time that could otherwise be utilized for various meaningful purposes. In Arpino and Bordone [42], grandparents who took care of their grandchildren were found to have a much lower rate of participation in social activities than those who did not care for their grandchildren, which could reduce their social capability to interact and entertain, thereby becoming prone to depression.

### 1.3. The Effect of Grandparenting on Life Satisfaction

On the one hand, life satisfaction among grandparents has been found to be associated with the intensity of grandparental care they provide. South Korean grandmothers who provide care in the long term tend to have better life satisfaction and self-rated health [20], as individuals obtain more self-esteem from carrying out multiple social roles [35]. Across the 10-year reference time frame, Lou (2011) [43] adopted a multistage sampling method to interview older adults in Hong Kong with grandchildren older than 12 and found that a reduced frequency with which the grandmothers played grandparenting roles was negatively associated with levels of life satisfaction. A similar outcome was found in Latin America: Chilean grandfathers who provided four or more hours per week of help reported better life satisfaction, and grandmothers who did the same were at a lower risk of depression [37].

On the other hand, grandparental care will also give rise to either a positive or a negative affective response right from the beginning and for the duration, during which the grandparents no longer have many choices once they have taken up the role of caregiver to their grandchildren. The positive sense of genealogical renewal and generational continuity may alleviate any negative effects. In addition, the extent of feelings about the importance of being a grandparent and of family continuity differ by the grandparents’ previous occupations. Song et al. [44] showed that grandmothers who had had more wealth and had worked in prestigious occupations such as emotional supporters and educators were more likely to show “generativity”, which was positively correlated with their life satisfaction. Active engagement with grandchildren was more fulfilling to grandmothers and positively related to their life satisfaction [45]. Chen et al. [46] have also found that the life satisfaction of the older ethnic minorities, relative to others, is much more dependent on intra-familial support.

### 1.4. The Current Study

A tremendous number of relevant studies on grandparental well-being from different perspectives internationally. Existing studies have indicated that grandparents living with their grandchildren generally enjoy an even better state of mental health and life satisfaction than those living in single-generation households [47]. Given the differences in cultural values attached to multigenerational co-residence between the East and West, grandparental care in the Chinese context may have different well-being consequences. However, empirical research on the impact of grandparenting on depression and life satisfaction in China remains scarce, and the evidence supporting the results is generally indirect [10]. To fill this research gap, this study empirically investigates the effect of grandparenting on depression and life satisfaction of middle-aged and older adults in China. We examined the association between co-residential grandparental care, depression, and life satisfaction, and then, among adults who participated in grandparenting, we estimated the extent to which the intensity of grandparental care, the number of grandchildren cared for, and the living arrangements determine adults’ depression and life satisfaction.

## 2. Data and Methods

### 2.1. Sample

This study used survey data from CHARLS 2018, a national longitudinal survey conducted by Peking University, which contained a sample of 19,816 individuals. CHARLS 2018 collected a set of micro-data representing families and individuals aged 45 or above in China, including demographic background, family information, family transfer, health status and function, work and retirement status, household income, and basic community conditions. Based on the population of districts/counties in 2018, using region (i.e., urban and rural areas) and GDP as stratified indicators, employing the probability proportional to size (PPS) sampling method, 150 districts/counties were randomly selected from 30 provinces across mainland China (Tibet was the only province not included). For each district/county, three villages/communities were selected according to PPS sampling. CHARLS designed and developed special mapping software (CHARLS-GIS) to perform on-site mapping and collect household information. Eighty households were randomly selected in each sample village/community, and the interviewers randomly selected a family member older than 45 years old as the main respondent, with his/her spouse also being interviewed. Given that our study focused on the relationship between taking care of grandchildren and well-being among older Chinese adults, all respondents without grandchildren were excluded from the analysis. Among those who had grandchildren, 7876 respondents completed the depression-related items, and 9118 respondents answered the question about life satisfaction. Samples with missing values for the explanatory variables and control variables were not analyzed, and the final samples consisted of 7778 respondents for the depression question and 8976 respondents for the life satisfaction question. To alleviate the effect of outliers, continuous variables, such as age, education, and income, were winsorized at the 1% level. The characteristics of all variables are presented in Table 1.

### 2.2. Dependent Variables

*Depression* was measured using a shortened version of the Center for Epidemiologic Studies-Depression Scale (CES-D) developed by Radloff [48] in 1977 with 10 items. The reliability and validity of the 10-item short form of the CES-D (CESD-10) among older Chinese adults have been shown to be satisfactory [49]. The frequencies of “I feel bothered by things”, “I have trouble keeping my mind on things”, and “I have felt depressed”, among others, were investigated. The options were coded as “0” for rarely or never (<1 day), “1” for some or a little (1–2 days), “2” for occasionally or a moderate amount of time, and “3” for most of the time (5–7 days). The scores were summed as the measurement of depression (range: 0–30); the higher the score, the greater the depressive symptoms.

*Life satisfaction* was measured using the question, “How satisfied are you with your life as a whole?” The options were coded using a 5-point Likert scale (ranging from 0 “not at all satisfied” to 4 “very satisfied”).

### 2.3. Explanatory Variables

*Grandparenting*. Among those who had grandchildren, CHARLS 2018 asked the respondents the question, “During the last year, did you/your spouse spend time taking care of your grandchildren?” The options were coded as 1 if the answer was “Yes” and 0 if the answer was “No”. For those who took care of a grandchild, CHARLS 2018 asked the respondents which children’s children they took care of, whether they lived together with these children, and how long they spent taking care of their grandchildren. Based on this information, we constructed three explanatory variables, *Grandparenting* (0 “not taking care of grandchildren”, 1 “taking care of grandchildren but not living with children”, and 2 “taking care of grandchildren and living with children”), *Time spent on care* (months), and *number of grandchildren* they took care of.

The analysis considered several demographic and socioeconomic variables as control variables. The respondents’ *gender* (0 = female, 1 = male), *age* (ranging from 46 to 89), *marital status* (0 = unmarried, 1 = married), *years of education* (ranging from 0 to 15), *ethnic minority status* (0 = no, 1 = yes), *hukou status* (0 = rural, 1 = urban), *religious beliefs* (0 = no, 1 = yes), *monthly income* (logarithm), and whether they were engaged in *non-agricultural work* (0 = no, 1 = yes) were included as covariates in the models.

### 2.4. Analytical Approach

We first present the characteristics of the variables by grandparenting status. Chi-square tests and Student’s *t*-tests were used to assess the differences between the categorical and continuous variables, respectively. Given that CHARLS 2018 used stratified multistage sampling and that individuals were nested in the communities, we used multilevel regression models. Compared with single-level multivariable analyses, multilevel regression can take the variability of different levels into account at the same time and estimate more accurate standard errors. All statistical analyses were carried out using STATA 15.0.

## 3. Results

### 3.1. Descriptive Statistics

Table 1 shows the characteristics of each variable (mean and standard deviation (SD) for the continuous variables, frequency, and percentage for the categorical variables) and difference test results between the grandparent and non-grandparent groups. The means for *depression* in the grandparent and non-grandparent groups were 8.62 and 9.46, respectively. There were no significant differences in *life satisfaction* between the grandparents and the non-grandparents. Among the respondents, 53.64% were women, and the mean age of the respondents was 64.85 (SD = 10.09). In addition, 26.90% of the respondents were separated, divorced, or widowed. Among those who had taken care of grandchildren over the last year, the mean *time spent taking care of grandchildren* was 3.29 months (SD = 3.49), and the mean *number of grandchildren* they took care of was 1.21 (SD = 0.52). About 59% of the respondents lived with their children while taking care of their grandchildren.

### 3.2. Association between Grandparenting, Depression, and Life Satisfaction

Analysis of variance component models showed that communities accounted for a considerable proportion of the total variance. The intra-class correlation coefficients (ICCs) for *depression* and *life satisfaction* were 0.08 and 0.04, respectively. Table 2 presents the results of the multilevel regressions addressing *depression* and *life satisfaction* among adults. Compared with those who did not take care of grandchildren last year, the respondents who took care of their grandchildren and lived with their children experienced fewer feelings of depression and were more satisfied with their lives, and these effects reached a statistically significant level (β = −0.956, *p* < 0.001; β = 0.077, *p* < 0.001, respectively). For those who took care of their grandchildren but did not live with their children, there were no significant differences in *depression* and *life satisfaction* compared with those who did not take care of their grandchildren.

The covariates selected showed associations with *depression* and *life satisfaction* in some way. The respondents who were male, older, married, well educated, with urban hukou status and higher monthly income, and who were engaged in non-agricultural work reported lower frequencies of feeling depressed. As for *life satisfaction*, the respondents who were older, married, had non-ethnic minority status, and were engaged in non-agricultural work tended to be more satisfied with their lives.

We further analyzed the impact of more indicators of taking care of grandchildren over the last year on the well-being of adults. The results are shown in Table 3. The association between *time spent taking care of grandchildren* and *depression* was negative and reached a statistically significant level (β = 0.098, *p* < 0.001). Living with children significantly reduced depression among adults who took care of their grandchildren (β = −0.731, *p* < 0.001). Meanwhile, spending more time taking care of grandchildren lowered life satisfaction (β = −0.009, *p* < 0.001), but taking care of more grandchildren was related to higher life satisfaction (β = 0.068, *p* < 0.001). Whether they lived with their children did not have a significant effect on life satisfaction among adults who took care of their grandchildren.

## 4. Discussion 

This study focused on two themes: first, it examined the association between co-residential grandparental care, depression, and life satisfaction; second, it studied the extent to which the intensity of grandparental care, the number of grandchildren cared for, and the living arrangements determine adults’ depression and life satisfaction. The results of this study can contribute to the diverse findings regarding the association between grandparenting, depression, and life satisfaction. As other scholars have previously noted, grandparenting is a cultural practice in which the arrangement of co-residential grandparenting is culturally expected and valued without any global gold standard [50]. This indicates that middle-aged and older Chinese adults with a more culturally appropriate approach to co-residential childrearing may lower their chances of experiencing depression and improve their life satisfaction compared with those who do not undertake grandparental care. This finding is similar to those of previous studies on Chinese grandparents, which showed that co-residential grandparenting causes grandparents to enjoy a better state of well-being than others [1,10,47]. Therefore, in Chinese metropolitan areas, communities can be built beyond the nuclear family structure to facilitate co-residential grandparenting, in which an elder care system, such as providing national healthcare packages regardless of hukou status [51], could be incorporated to enable adults from rural areas to co-reside or live in close proximity to their children and grandchildren. This could improve the mental health of adults through intra-familial intergenerational emotional exchanges while providing a safety net for their physical well-being, should they need hospital treatment or medication. The government could also institutionalize pilot initiatives such as community-based elder care, which have previously been used to mitigate the shortages of nursing staff in metropolitan areas and provide healthcare, food, and other assistance within the communities [52]. This would further help adults whose children live in cities to co-reside with their grandchildren without passing burdens to their adult children, who now need to pay back the grandparental care received with their parents’ healthcare or caring needs.

In this study, we also examined how the intensity of grandparenting, in terms of time spent on childrearing and the number of grandchildren cared for, and co-residential status affect adults’ depression and life satisfaction. Our results suggest that the more time adults spend on childrearing, the more likely they are to suffer from depression and to have lower life satisfaction. Seen from the “role strain” perspective of role theory, when people’s physical and psychological resources for well-being cannot cope with the intensity of their role, mental status and life satisfaction will be negatively affected [53]. Additionally, in previous studies, Chinese grandparents who undertake heavy childcare beyond their capabilities have been found to have much lower cognitive functions and accelerated health decline over time [1,54], in turn affecting their mental depression and life satisfaction. Therefore, it is important that nudging policies be implemented to encourage parental involvement in childcare to reduce the burden on older adults. This would allow grandparental care to take place at a low intensity to promote the well-being of adults.

Studying the experience of American grandmothers who were the primary caregiver for one or more grandchildren, Dowdell [55] found that the number of grandchildren needing care had a negative impact on adults’ schedules and on the perceived adequacy of family support, which affected their well-being. However, our study of Chinese grandparents does not support this qualitative finding. The results suggest that the more grandchildren the adults take care of, the higher their life satisfaction and the lower their mental depression will be, after controlling for caring intensity and living arrangements. The basic reason why the results are different is probably related to the cultural differences between grandparents in different parts of the world. In the United States and European nations, the motivations behind grandparenting tend to be related to economic reasons or to the marital instability of adult children, whereas, in China, grandparents see grandparenting as their responsibility and part of the moral code of conduct to obtain fulfillment and life satisfaction [56,57]. When adults simply have more children to take care of, controlling for the time and effort spent on childrearing, they would not see them as extra burdens but as “virtues” according to Chinese Confucian culture [58]. That is, Chinese culture facilitates a strong role-enhancement effect, as grandparents caring for more grandchildren fulfill their socially and culturally expected roles [54,59]. Additionally, unlike Dowdell [55], our study included grandparents who were not the primary caregivers but offered subsidiary family education to grandchildren beyond that of their parents. Therefore, in our study, with the help of adult children, the number of grandchildren cared for may not translate directly into the intensity of grandparental care, and thus, the role-enhancement effect associated with caring for more children may outweigh the role strain effect associated with caring intensity. In China, the recent changes that have gradually abolished the historically harsh limitations on the number of children per couple could further boost the well-being of adults, as they may have the opportunity to care for more grandchildren. However, relevant policies should be in place to enable adult children to alleviate the intensity of the burden of grandparental care.

Meanwhile, co-residing with grandchildren may be a further role enhancement for older Chinese adults as they fulfill the cultural and social expectations of Confucian culture [54]. As shown in our study, adults who co-resided with their grandchildren experienced significantly lower levels of depression and higher levels of life satisfaction compared with those who did not, while holding other variables constant. This result suggests that the living arrangement is an important external factor that determines the impact of grandparenting on adults’ well-being, which is consistent with social exchange theory, given that when co-residing with adult children, the grandparental care could largely alleviate the caring burden from adult children, who, in return, may provide emotional and economic support. Through this intergenerational exchange, middle-aged and older people enjoy better mental health and life satisfaction. Indeed, this finding may conflict with previous studies that have suggested that living in skipped-generation households is negatively associated with happiness among grandparents when only controlling for gender, education, and income [60]. However, as grandparents co-residing with their adult children are also the most likely to undertake high-intensity grandparental care [61], the results of the study by Wen et al. [60], which failed to control for the factors related to caring intensity or the number of grandchildren, may not be accurate. Based on our findings, we believe that social policies that promote residential proximity between grandparents and their children are crucial for maintaining their mental well-being and life satisfaction. Meanwhile, guidance and recommendations from the government, communities, and employers also need to be in place to promote parental involvement in child care and protect the co-residing older adults from engaging in high-intensity grandparental care.

The limitations of this study are worthy of discussion and may provide directions for further research in this field. First, in this study, the causal relationship between grandparental care and well-being was assumed to be unidirectional. Second, the analysis relied on using the subset of respondents who had grandchildren from CHARLS 2018. The older adults involved in grandparental care naturally needed to have better mental and physical well-being to meet the demands of daily grandparenting, which may contribute to endogeneity in the correlation studied. For future studies on the older adults involved in grandparental care, we will need to explain the nature of self-selection-based endogeneity bias.

## 5. Conclusions

This study examines the association between grandparenting, depression, and life satisfaction among middle-aged and older adults in the context of Chinese culture. Furthermore, it studies the extent to which the intensity of grandparenting, the number of grandchildren cared for, and the living arrangements determine adults’ depression and life satisfaction. The results indicate that co-residential grand parental care might reduce depressive symptoms and improve the life satisfaction of middle-aged and older adults in China, compared to those who do not take care of grandchildren and those who take care of grandchildren but live separately from children. Among those who take care of grandchildren, the more time that adults spend on childrearing, the more likely they are to suffer from depression and to have lower life satisfaction. Meanwhile, taking care of more grandchildren is related to higher level of life satisfaction and living with children is related to less depressive symptoms. Our findings could form a basis for the development of effective policy intervention for policymakers to improve the quality of life of Chinese adults. 

## Figures and Tables

**Table 1 ijerph-19-10790-t001:** Characteristics of the variables, by grandparenting status.

	Full Sample		Grandparenting		Non-Grandparenting		*p*
	Mean (Frequency)	SD (%)	Mean (Frequency)	SD (%)	Mean (Frequency)	SD (%)	
** *Dependent variables* **							
Depression	9.04	6.69	8.62	6.37	9.46	6.98	<0.001
Life satisfaction	2.23	0.83	2.24	0.81	2.22	0.85	0.269
** *Explanatory variables* **							
Time spent on care (month)	1.52	2.89	3.29	3.49			
Number of grandchildren taken care of	1.21	0.52	1.21	0.52			
Living with children							
Yes	2658	27.61	2658	59.52			
No	6969	72.39	1808	40.48			
** *Control variables* **							
Gender							0.571
Female	5160	53.64	2379	53.33	2781	53.91	
Male	4460	46.36	2082	46.67	2378	46.09	
Age	64.85	10.09	61.02	7.89	68.16	10.61	<0.001
Marital status							<0.001
Married	7032	73.10	3693	82.78	3339	64.72	
Unmarried	2588	26.90	768	17.22	1820	35.28	
Years of education	5.06	4.06	5.71	4.02	4.51	4.01	<0.001
Ethnic minority							0.023
Yes	734	7.63	370	8.29	364	7.06	
No	8886	92.37	4091	91.71	4795	92.94	
Hukou							0.032
Rural	7736	80.79	3546	79.86	4190	81.60	
Urban	1839	19.21	894	20.14	945	18.40	
Religious beliefs							0.038
Yes	1056	10.98	458	10.27	598	11.59	
No	8564	89.02	4003	89.73	4561	88.41	
Monthly income (ln)	4.24	2.95	4.00	3.15	4.45	2.74	<0.001
Non-agricultural work							<0.001
Yes	1938	20.18	1066	23.93	872	16.94	
No	7666	79.82	3389	76.07	4277	83.06	

Note. The *p*-values were produced by a chi-square test for the categorical variables and by Student *t*-tests for the continuous variables.

**Table 2 ijerph-19-10790-t002:** Multilevel regression of grandparenting on depression and life satisfaction among Chinese adults.

	Depression		Life Satisfaction	
Variables	Coef.	Robust SE	Coef.	Robust SE
** *Explanatory variables* **				
Grandparenting (No = ref.)				
Grandparenting, not living with children	−0.292	0.194	0.017	0.024
Grandparenting, living with children	−0.956 ***	−0.174	0.077 ***	0.021
** *Control variables* **				
Gender (Female = ref.)	−1.615 ***	−0.161	0.033	0.020
Age	−0.031 **	−0.010	0.008 ***	0.001
Marital status (Unmarried = ref.)	−1.154 ***	−0.189	0.070 **	0.023
Ethnic minority (No = ref.)	−0.048	−0.325	0.104 **	0.038
Educational years	−0.222 ***	−0.021	−0.001	0.003
Hukou (Rural = ref.)	−0.874 ***	−0.227	0.025	0.027
Religious believer (No = ref.)	−0.240	0.249	0.022	0.030
Monthly income (ln)	−0.119 ***	0.029	0.006	0.004
Non-agricultural work (No = ref.)	−1.003 ***	0.198	0.094 ***	0.025
Constant	16.067 ***	0.750	1.550 ***	0.090
** *Model statistics* **				
** *n* **	7778		8976	
Log-likelihood	−25,363.614		−10,960.212	
AIC	50,755.23		21,948.42	
BIC	50,852.65		22,047.86	

Note. Ref: reference group; AIC: Akaike information criterion; BIC: Bayesian information criterion; ** *p* < 0.01; *** *p* < 0.001.

**Table 3 ijerph-19-10790-t003:** Multilevel regression among Chinese adults who took care of grandchildren last year.

	Depression		Life Satisfaction	
Variables	Coef.	Robust SE	Coef.	Robust SE
** *Explanatory variables* **				
Time spent on care (months)	0.098 ***	0.030	−0.009 *	0.004
Number of grandchildren taken care of	−0.109	0.199	0.068 **	0.025
Living with children (No = ref.)	−0.731 ***	0.203	0.047	0.026
** *Control variables* **				
Gender (Female = ref.)	−0.845 ***	0.222	−0.008	0.028
Age	−0.021	0.016	0.005 *	0.002
Marital status (Unmarried = ref.)	−1.279 ***	0.289	0.042	0.036
Ethnic minority (No = ref.)	−0.002	0.414	0.107 *	0.050
Years of education	−0.237 ***	0.029	0.004	0.004
Hukou (Rural = ref.)	−0.917 **	0.294	−0.003	0.037
Religious believer (No = ref.)	−0.040	0.344	0.010	0.043
Monthly income (ln)	−0.123 **	0.038	0.005	0.005
Non-agricultural work (No = ref.)	−0.854 ***	0.258	0.057	0.033
Constant	14.873 ***	1.102	1.767 ***	0.138
** *Model statistics* **				
** *n* **	3891		4327	
Log-likelihood	−12,510.52		−5180.327	
AIC	25,051.04		10,390.65	
BIC	25,145.03		10,486.24	

Note. Ref: reference group; AIC: Akaike information criterion; BIC: Bayesian information criterion; * *p* < 0.05; ** *p* < 0.01; *** *p* < 0.001.

## Data Availability

The raw data supporting the conclusions of this article will be made available by the corresponding author, without undue reservation.

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
