# Peer review of "The Effect of Grandparenting on the Depression and Life Satisfaction among Middle-Aged and Older Chinese Adults"

_ijerph, 2022, doi:10.3390/ijerph191710790_

Round 1

Reviewer 1 Report

 The research, which deals with the contribution of grandparenting and its various components to the well-being of older Chinese adults, is interesting and can make an important contribution to the body of research knowledge on this subject, as well as constituting a basis for the development of appropriate ageing policies and services. In addition, the focus on the unique cultural characteristics of China in relation to grandparenting is relevant and deepens and enriches the research. However, the research in its current form is not sufficiently coherent and focused.

1.      The introduction sub-chapter is very long, and the difference between it and the literature review is not clear enough. For example, studies regarding grandparenthood were reviewed both in the introduction and in the literature review. Moreover, while in the introduction, sub-chapter there is a focus on cultural differences between East and West grandparenthood, in the literature review chapter, these differences tend to be blurred.

2.      The aim of the study, which appears at the end of the introduction sub-chapter, requires more clarification: What is the meaning of ''examined differently''? ("The study examined how grand-parenthood affects the well-being of Chinese adults differently, while controlling for their preliminary health-related circumstances."). In the first paragraph of the discussion chapter, the aim of the study is formulated more clearly (page 10).

3.      The literature review is extensive and covers relevant topics. However, it is not organized enough. There is repetition, overload, and lack of clarity. It is recommended to organize the information in an orderly manner, with a clear and orderly reference to the research variables (e.g., time spent on care, living with children etc.).

4.       Many of the studies cited in the literature review chapter are from the 1990s and some even from the 1980s. It is recommended to bring more up-to-date research from the last decade.

5.      Grandparenting as a process of counter-transition is mentioned twice:  in the introduction sub-chapter (page 1, last paragraph) and in the sub-chapter titled "Grandparenting in China" (page 2, first paragraph). However, the references are different (Arpino et al., 2018 and Hagestad, 1985). The difference between the two citations should be clarified.  

6.      In the sub-chapter titled "The Influence of Grandparenting on Well-being among Older Adults," the difference between subjective and objective indicators of well-being is not clear enough.

7.      In the sub-chapter titled "The Influence of Grandparenting on Well-being among Older Adults," the last sentence is not clear (e.g., "As subjective well-being can also be measured by objective indicators, these two distinct perspectives should not be confused with their measurements" (Sumner, 1996)).

8.      The connection of the claim that "Less-educated individuals may be more concerned with family identities than middle-class individuals…" to the literature review is not clear enough.

9.      The theory of role strain, which appears in the discussion section, is important and relevant (page 10, second paragraph). It is recommended to present this theory in the literature review chapter.

10.   In the discussion chapter, new and relevant studies on grandparenting are presented. It is recommended to present them in the introduction chapter, in lieu of the outdated studies from the 1990s and the 1980s.

Reviewer 2 Report

A well-written and presented study on the relationship between grandparenting and well-being among Chinese adults using multilevel regression models. I have no further comments, but to congratulate the authors.  

Author Response

We are very grateful for your endorsement of our research, which gives us more confidence in this study.​

Reviewer 3 Report

Dear authors, congratulations on your work. In my opinion, this paper, before publication, should solve the following points:

Title: should be more specific because well-being is much more than depression and life satisfaction. In this way, I suggest that the title includes the variables of the study and only not the well-being.

Introduction: This section is big...it is two times bigger than the discussion/conclusion. The introduction will be getting better with a reduction.

The subjective well-being definition should be better…the authors should talk about variable affective and cognitive well-being and justify why they only used life satisfaction and not the other variables, like affective variable.

The methodologic point is a big concern about this work. First, people 45 years old are not older. Second, the authors used instruments validated for older people, but the study sample has people under 60/65 years old?! In this way, the goal of this work should be changed to include this situation.

Round 2

Reviewer 1 Report

The manuscript is now clearer and tighter. However, despite the improvement in writing, there are still some significant issues in the literature review that should be addressed

1.      The introduction sub-section mainly focuses on grandparenting in East Asia and the differences between that and grandparenting in Western countries. However, the role of the introduction is to briefly present the rationale for the study and the research question. These aspects are not clear and emphasized enough in the introduction. The

2.      The sentence " On the one hand, grandparenting is often seen more as a challenge than a blessing in the West" (P.1, second paragraph) can be perceived as too harsh. Despite the studies that emphasized the burden of grandparenting in the West, it is not certain that the burden still overshadows the blessing of grandparenting, in Western countries. It is recommended to soften this statement and present a more complex and less unequivocal picture of the experience of grandparenting in the West

3.      The use of the word "influence" in the title of the manuscript, as well as in the titles of the sub-sections in the literature review section is problematic. It is recommended to use the words "effect" or "contribution"

Author Response

Thanks very much for your interest in our paper and giving us the precious opportunity to revise it. We have benefited a lot from your valuable and constructive suggestions.

Below, we present a point-by-point summary of our responses to the questions and comments. Please note that “Q” means the question and “A” means our response to the question.

Q1. The introduction sub-section mainly focuses on grandparenting in East Asia and the differences between that and grandparenting in Western countries. However, the role of the introduction is to briefly present the rationale for the study and the research question. These aspects are not clear and emphasized enough in the introduction.

A: Thanks for the suggestion. In the revised manuscript, we have added additional discussions on the research rationales and the research questions focusing on the effect of grandparenting on depression and life satisfaction in China in the last subsection of Introduction.

Q2. The sentence " On the one hand, grandparenting is often seen more as a challenge than a blessing in the West" (P.1, second paragraph) can be perceived as too harsh. Despite the studies that emphasized the burden of grandparenting in the West, it is not certain that the burden still overshadows the blessing of grandparenting, in Western countries. It is recommended to soften this statement and present a more complex and less unequivocal picture of the experience of grandparenting in the West

A: Thanks for the suggestion. In the previous version, the description of Western grandparental care was indeed too harsh. In the revised manuscript, we have rephrased the discussion by admitting that there are both blessings and challenges brought by grandparental care in the West.

 Q3. The use of the word "influence" in the title of the manuscript, as well as in the titles of the sub-sections in the literature review section is problematic. It is recommended to use the words "effect" or "contribution"

A: Thanks for the suggestion. We understand the term “influence” may create confusion. Hence, we have replaced “influence” by “effect” or “affect” accordingly.

Reviewer 3 Report

Dear authors, thank you for addressing my comments.

Congratulations on your work.

Author Response

Thanks very much for your interest in our paper and giving us the precious opportunity to revise it. We have benefited a lot from your valuable and constructive suggestions.